# A systematic review on nomophobia prevalence: Surfacing results and standard guidelines for future research

**Ana C. León-Mejía**[1][*], **Mónica Gutiérrez-Ortega**[1], **Isabel Serrano-Pintado**[2], **Joaquín González-Cabrera**[1]

**1** Department of Psychology, Faculty of Education, Universidad Internacional de La Rioja (UNIR), Madrid, Spain, **2** Department of Psychology, Faculty of Psychology, Universidad de Salamanca, Salamanca, Spain

☯ These authors contributed equally to this work.

\* aleon@unir.net

## Abstract

### Background

Mobile phones allow us to stay connected with others and provide us a sense of security. We can work, chat with family and friends, take pictures, buy clothes or books, and even control home appliances. They play such a significant role in our lives that we feel anxious without them. In some cases, the relationship between humans and these communication devices have become problematic. Nomophobia (NMP) is the fear of becoming incommunicable, separated from the mobile phone and losing connection to the Internet. Since this social phobia was coined in the first decade of the XXI century, a growing number of studies have studied it and reported the prevalence of this technology-related problem. However, this research activity has generated mixed results regarding how we assess and report nomophobia and who may be at a higher risk of suffering or developing it.

### Methods

We conducted a systematic review of 108 studies published in English and Spanish and collected them in Parsifal. We searched for assessment and prevalence data on nomophobia. Also, we looked at gender and age differences to identify risk factors and see if these differences exist and emerge worldwide.

### Results

In this study, we find that women and younger individuals suffer more from nomophobia. The disparity in reporting the prevalence of nomophobia is enormous since the percentages of "at-risk" participants go from 13% to 79%, and participants suffering from it are between 6% and 73%, being the score in the range of 45.5 and 93.82. Within the group of nomophobic people, moderate cases vary between 25.7% and 73.3%, and severe cases, between 1% and 87%. Such disparity is due to differences in assessment criteria. Females and young people seem to be more vulnerable to nomophobia although methodological disparity

**Data Availability Statement:** All relevant data are within the manuscript.

**Funding:** This research was funded by the Spanish Ministry of Economy, Industry and

Competitiveness, RTI2018-094212-B-I00: (CIBER-AACC), and PID2019-107589GB-I00, and it was also supported by the International University of La Rioja, Project Cyberpsychology (Triennium 2017-2020).

**Competing interests:** The authors have declared that no competing interests exist.

makes it difficult to reach definitive conclusions. We conclude our review by recommending some common guidelines for guiding future research.

## Introduction

Nomophobia (No-Mobile-Phone Phobia) is a modern-day problem that was first coined by the UK Post Office in 2008. The Post Office ordered a research organization called YouGov to examine whether phone users in the UK were experiencing anxiety over their mobile phones. It was found that almost 13 million people reported being anxious when losing their mobile or forgetting to take the phone with them, running out of battery, having no network coverage, and when not receiving any calls, texts or emails for some time, which represents a 53% of the surveyed [1].

In this way, society was ahead of science in identifying a phenomenon that was raising social concern: the relationship with our mobiles and the alleg problems created by this technological link. Let us remember that mobiles appeared early in the seventies. And since then mobile connections (8.97 billion) have surpassed the number of people in the world (7.8 billion), becoming the fastest-growing human-made technology that has ever existed [2]. These devices are continuously evolving to be more attractive, compelling, and faster, and mobile companies are also competing to offer us new models with more memory, better cameras, and batteries, while the number of apps and services are also constantly increasing, making us dependent on them.

We have named a problem—nomophobia—but we are just beginning to understand why people experience anxiety when being out of touch or why they never want to turn their mobile off, and why our mobiles are the first thing to check in the morning and the latest at night. In order to answer all these questions, we need to understand this phenomenon better. King [3, 4], and Yildirim [5] were the first scholars to address this task. In the study by King et al. [3], nomophobia was regarded as a 21st-century disorder resulting from information and communication technologies. They posited that nomophobia comes from the fear of not being able to communicate with others and being separated from the mobile or not connected to the Internet. In another work, King et al. [4] spoke of nomophobia as a situational phobia characterized by a fear of becoming distressed and not getting any assistance. Yildirim [5] operationalized this theoretical construct into a research instrument consisting of a self-reported measure (the Nomophobia Questionnaire, NMP-Q) that examines our relationship with smartphones, i.e., mobiles with internet connections that run software programs in a way similar to a computer. In doing so, nomophobia was linked to a problematic mobile phone use, defined as an incapacity to control and regulate the use of the mobile phone and suffering negative daily life consequences. After the NMP-Q was developed and proved to be a valid instrument to assess this problem, many scholars have translated and adapted it to other languages, including Spanish, Chinese, Italian, Persian, and Indonesian, among others. With these new versions, we also started to have more prevalence data and other valuable worldwide information regarding the profile of people who suffer from nomophobia.

Rodríguez-García et al. showed in their systematic review on a sample of 42 studies [6] that since 2010 (when scholars first started to talk about nomophobia) this problem has been studied regarding a growing number of psychological variables such as anxiety, panic disorder, stress, depression, obsessiveness, FOMO (Fear of Missing Out), extraversion, awareness, emotional stability, sympathy, openness to experience, mindfulness, loneliness, and self-happiness,

among others; they also review the connections between nomophobia and Internet usage, social media, academic performance, learning and attention, and collectivism explored by the literature. Their results highlighted that most of the research conducted so far was exploratory. We would add to these conclusions that nomophobia itself is not still well known, as substantial dark spots are hampering our understanding. For instance, terms such as addiction or dependence are frequently used interchangeably with nomophobia, both in the academic literature and in colloquial conversations, creating conceptual confusion. As for who may be at a higher risk of developing nomophobia, data is non-conclusive because some studies report that women are more at risk than men, while others state the opposite; the same happens when examining age differences. Even more importantly, there are many methodological gaps regarding nomophobia assessments and prevalence reporting that we need to address. None of these questions have been studied to date.

For this reason, and to clarify some points of significant concern in the nomophobia literature, we have conducted a systematic review of nomophobia prevalence and gender and age differences. Given the disparity of criteria used to study and report nomophobia, we also aimed at developing some standard guidelines to help us study, compare, and systematize future research results.

## Methods

### Protocol and eligibility criteria

The number of studies on nomophobia has grown remarkably since this concept was first coined in 2008 and, mainly, since the NMP-Q came out [5, 7]. A myriad of results and definitions have arising several significant research questions, among which we are going to focus on the following:

- Do we similarly measure nomophobia?

- Are there gender and age differences in nomophobia prevalence?

- Do we similarly report nomophobia prevalence?

- How should nomophobia be reported?

To answer these research questions, we conducted a systematic review of 108 papers published up to the 1st of January 2020. We followed the Preferred Reporting Items for Systematic Reviews and Meta-Analyses (PRISMA) guidelines [8], as well as the specific psychology and health considerations suggested by Perestelo-Pérez [9] and by Shamseerg et al. [10]. First, papers were included in this systematic review on three starting conditions: a) If they were written in English or Spanish; b) If they were peer-reviewed; c) If they were indexed in electronic academic databases and search engines.

Second, we only considered those papers that focused on nomophobia from a psychological and applied perspective with quantitative data on prevalence. Interpretative and therapy studies were excluded except for the work of Yildirim [11] that was part of a mix-method research that led to the NMP-Q development.

Third, studies aiming at mobile addiction and dependence were eliminated from the study as these are related but different constructs. Sometimes the word nomophobia was in the title and abstract, but the conceptual frame and research tool focused on mobile addiction or other topics unrelated to nomophobia, and thus we omitted those papers too.

Also, we excluded studies whose results were highly unclear, precluding us from understanding what was reported, or that they did not provide at least one of the following pieces of information: which instrument was used, NMP prevalence, and gender/age differences. We

omitted theses (except for Yildirim's thesis developing the NMP-Q), master's theses, and non-scientific publications.

The correct handling of duplicate studies is a critical issue for a systematic review and, therefore, we followed the recommendations of Kwon et al. [12] and decided to eliminate them one by one in *Parsifal* (where all of them were collected) after a careful checking by two members of the research group (MG and AL).

## Search strategy

Relevant articles were identified by searching *Google Scholar*, *Web of Science*, *Scopus*, *ProQuest*, and *Science Direct* and gathered in *Parsifal.* The first search took place in May 2018, filling in the following search fields: title, keywords, and abstracts. A second search was done in May 2019 and December of 2019 to update the study.

We used the following terms searched by Boolean operators (*and, or)* and truncations and wildcards (*? \**): *nomophobia, nomophobes, nomophobic, nomofobia, nomofóbico/a.*

Two researchers (AL, MG) performed an eligibility assessment in an unblinded, standardized, and independent manner. Firstly, the title, keywords, and abstract of the papers initially found were screened to eliminate those not matching initial inclusion criteria (language and peer review). Secondly, AL and MG reviewed the full-text of those articles that were eligible for inclusion. Disagreements on inclusion/exclusion of selected articles were solved after four researchers met and discussed its eligibility, reaching a 100% agreement afterward. Articles not meeting the inclusion criteria were removed from the study (see Fig 1).

## Risk of bias

No doubt, the existence of bias in systematic review (SR) is almost inevitable. Even though we tried to minimize it, there can always be some subjectivity in the screening process. In this SR, the two principal authors, who did the analyses, were in charge of this task and agreed 100% on the studies finally included, a process that is known to reduce the risk of bias considerably but not eliminate it [13].

Searching institutional websites is crucial to avoid publication bias, as research gathered in these repositories may contain relevant information [14, 15]. But it also has been said that these sources of information introduce other types of bias; for instance, differences in search functions across websites make it necessary to change or adapt the search strings [13]. Also, not all repositories are equally visible on the Internet or accessible to the researcher, and most of the debate is around unpublished trials not being represented in the SR. This latter fact does not affect this SR since the literature on NMP that we reviewed is not based on controlled trials but mostly on prevalence assessment and correlational designs.

In any case, we tried to minimize bias by screening Google Scholar and other article repositories. Still, most of the publications on nomophobia that came out of the peer-reviewed channels did not meet all of our inclusion criteria (i.e., language, report prevalence/assessment data, and meeting a minimum of scientific standards of research). Those meeting our requirements were indeed included. Also, we analyzed a significant number of studies whose journals are not indexed in SCOPUS or WoS (mostly from India and Pakistan). Therefore, even though the possibility of missing pertinent studies is there, we followed all the steps to minimize it.

## Data extraction and qualitative analysis

Before analyzing the prevalence of nomophobia in-depth, it is worth exploring some general data about the studies that have been conducted until now, as this will give us a global picture

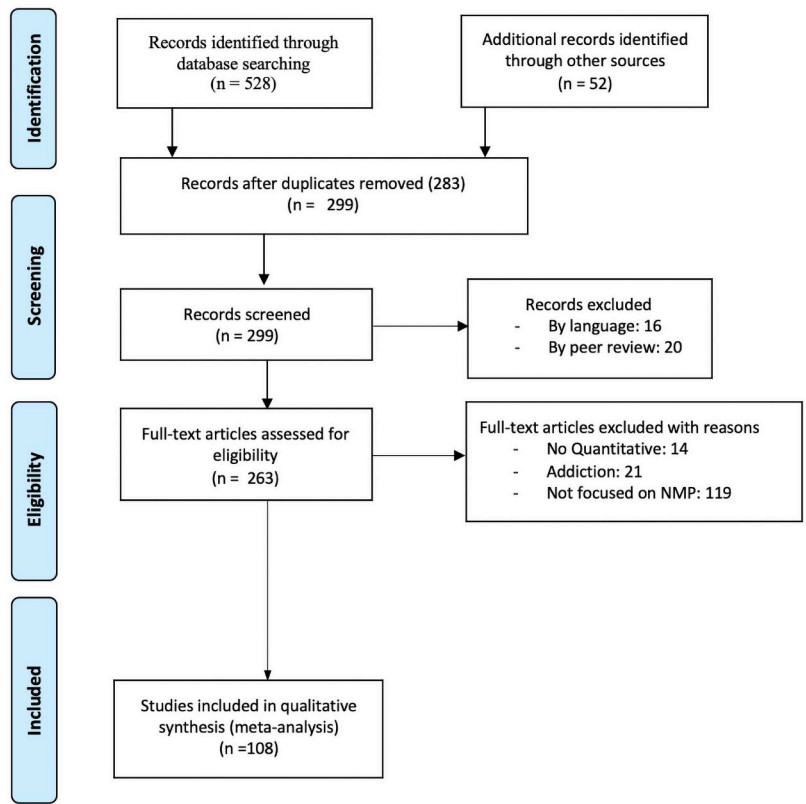

**Fig 1. PRISMA flow diagram.** Note: "*No Quantitative*" refers to lack of prevalence/assessment data and not to the research methodology.

of the nomophobia research. India and Turkey are the countries that have led the research in (29 and 28 papers, respectively), followed by USA (9) and Spain (8).

The publication rate started to rise in 2015, being 2017, 2018 and 2019 the years that concentrate most of the publications.

Most studies were done with students, particularly undergraduates in the field of health sciences (mostly Medicine and Nurse Studies) although not all studies reported this information or employed mixed samples (students and non-student). The sample size is very heterogeneous varying from less than 30 participants to more than 3200.

## Nomophobia research on its prevalence

The most used instrument is the NMP-Q (82), followed by ad hoc questionnaires or surveys (16), and other scales and inventories (10) shown in Table 1. To date, there are fifteen versions of the NMP-Q since the first English version was published between 2014 (Ph.D. version) and 2015 (final publication in a peer-review journal) by Yildirim and his collaborators.

In Spanish, we have three versions with some differences between them. The first version by Gutiérrez-Puertas et al. [16] has a tiny sample (n = 65), whereas the version of Ramos-Soler at al. [17] and González-Cabrera et al. [18] used more significant samples (372 and 306, respectively). The main difference between the latter two is the likert-scale used: while the study by Ramos-Soler et al. [17] changed the original scale reducing the responses from 7 to 5, the work of González-Cabrera et al. [18] did not change anything from the original version.

**Table 1. NMP assessment instruments.**

| NMP-Q | Year | Authors |
|---|---|---|
| **Version** | | |
| *NMP-Q (English)*, First version | 2014-2015 | Yildirim & Correia |
| *NMP-Q (Turkish)* | 2016 | Yildirim et al. |
| *NMP-Q (Spanish)* | 2016 | Gutiérrez-Puertas et al. |
| *NMP-Q (German)* | 2017 | Davie & Hilber |
| *NMP-Q (Spanish)* | 2017 | Ramos-Soler et al. |
| *NMP-Q (Spanish)* | 2017 | González-Cabrera et al. |
| *NMP-Q (Iranian)* | 2018 | Lin et al. |
| *NMP-Q (French)* | 2018 | Tams et al. |
| *NMP-Q (Persian)* | 2018 | Elyasi et al. |
| *NMP-Q (Arabic)* | 2018 | Al-Balhan et al. |
| *NMP-Q (Italian* | 2018 | Adawi et al. |
| *NMP-Q (Chinese* | 2018 | Jianling & Chang |
| *NMP-Q (Indonesian)* | 2018 | Rangka et al. |
| *NMP-Q (Tamil)* | 2018 | Mallya et al. |
| *NMP-Q (Portuguese)* | 2019 | Galhardo et al. |
| *Other questionnaires*\* | | |
| *NSI- SR* | 2013 | Bivin et al. |
| *MPUQ* | 2014 | King et al. |
| *TMD* | 2015 | Yildirim & Correia |
| *PUMP* | 2016 | Datta et al. |
| *RWT* | 2016 | Matoza & Carballo |
| *ICD-10 Criteria for Dependence syndrome* | 2017 | Dongre et al. |
| *IAT* | 2017 | King et al. |
| *ERA-RSI* | 2018 | Montserrat et al. |
| *CERM* | 2018 | Salinas et al. |
| *QANIP* | 2018 | Olivencia-Carriï¿½n et al. |

**Abbreviations**: *NSI-SR*: Nomophobia Severity Index-Self Related Version; *PMUQ*: Mobile Phone-Use Questionnaire; *TMD*: Test of Mobile Phone Dependence; *PUMP*: Problematic Use of Mobile Phones Scale; *RWT*: Robert Weiss Test; *ICD-10*: International Classification of Deseases (10th version: Criteria for Dependence Syndrome); *IAT*: Internet Addiction Test; *ERA-RSI*: Scale of Addiction-Adolescent Risk to Social Networks and Internet; *CERM*: Mobile Experience Questionnaire; *QANIP*: Questionnaire to Assess Nomophobia.

As for India, which is one of the leading countries in the number of publications using the NMP-Q, it is unknown whether they were using the English version or an Indian translation because this piece of information is not disclosed. However, the only publication of an Indian version *per se* is the adaptation to Tamil in 2018. In general, many studies neglected to inform which version of the NMP-Q was being employed; this information was taken for granted. The work of Yildirim [5] coined two words: *nomophobe* (someone who suffers from nomophobia) and *nomophobic* (the characteristics of nomophobes and/or behaviors related to nomophobia), but these words are not always used in the intended way. This may be due to the fact that "ic" is a common termination to convert nouns into adjectives, e.g., academia/academic. This may explain why *nomophobic* is more used than *nomophobe*. And for this reason we will also use "nomophobic" to name people having nomophobia.

Regarding the prevalence data gathered with the NMP-Q (see Table 2), the main obstacle for analyzing results is twofold. First, we find an enormous heterogeneity in assessing NMP.

**Table 2. Levels and classification systems with the NMP-Q.**

| CLASSIFICATION | STUDY | CRITERIA | RESULTS |
|---|---|---|---|
| Meeting one criterion<br>Scoring above average<br>Scoring above average<br>Scoring above average | Lee et al. (2017)<br>Adnan & Gezgin (2016)<br>Gezgin & Çakır (2016)<br>Gezgin et al. (2017a) | 61 or higher<br>NMP = being above average (4.07)<br>NMP = being above average (3.72)<br>NMP = being above average (3.96) | 54.35 ± 14.48<br>Participants are above average<br>Participants are above average<br>Participants are above average |
| 3-level CL | Asensio et al. (2018) | Mild: 21-48<br>Moderate: 49-77; Severe: 78-105 | **45.5 ± 12.6**<br>A 79.3% score between 35-60 |
| 3-level CL | Mallya et al. (2018) | Absence: <34<br>At risk: 34-39<br>NMP: >40 | Absence: 7.6%<br>At risk: 13%<br>*Severe: 86.9%* |
| 3-level CL | Deryakulu & Ursavaş (2019) | None to Mild: 20-59<br>Moderate: 60-99; Severe: 100-140 | 72.46 ± 22.96<br>Moderate: 56.8%; Severe: 13.6% |
| 3-level CL | Adawi et al. (2019) | Mild: 21-59<br>Moderate: 66-99; Severe: ≥100 | Mild: 51.1%<br>Moderate: 41.4%; Severe: 7.4% |
| 3-level CL | González-Cabrera et al. (2017) | 15P (Occasional): 39<br>80P (At risk): 87*<br>95P (Problematic user): 116 | 67.31± 25.7<br>P15: 14.4%<br>P80: 66.4%<br>P95: 4.6% |
| 4-level CL | Yildirim and Correia. (2015a) | Absence 0-20; Mild: >20<60<br>Moderate: >60<100 Severe:>100 | No prevalence results |
| 4-level CL | Nagpar & Kaur (2016) | Absence: <20; Mild: >20<60<br>Moderate: 60<100 Severe:>10 | *76.01 ± 14.98* |
| 4-level CL | Apak & Yaman (2019) | None, less, medium, high<br>(Kluster mean analysis) | None: 23.1%<br>Low: 35.8%<br>**Moderate: 25.7%** High: 15.3% |
| 4-level CL | Davie & Hilber (2017) | Absence: ≤20; Mild: 21-60<br>Moderate: 61-100; Severe: 101-120 | Mild: 57%<br>Moderate: 40%; Severe: 3% |
| 4-level CL | Gezgin et al. (2018b) | Absence: <20; Mild: 21-60<br>Moderate: 60-100; Severe:≥100 | 3.97 ± 1.37 |
| 4-level CL | Ayar et al. (2018) | Absence: 0-20; Mild: 21-59<br>Moderate: 60-99; Severe: 100-140 | Absence: 3%<br>Moderate: 51.9%; Severe: 13.6% |
| 4-level CL | Bartwal & Nath (2019) | Absence: <20; Mild: 20-60<br>Moderate: 60-100; Severe:≥ 100 | Mild: 15.5%<br>*Moderate: 67.2%*; Severe: 17.3% |
| 4-level CL | Yavuz et al. (2019) | Absence: <20 Mild: 21-59<br>Moderate: 60-99; Severe:≥100 | ♀: 70.52 ± 25.22: 64.23 ± 25.28<br>♀ Mild: 35%<br>♀ Moderate: 50% Severe: 14%<br>♂ Mild: 45%<br>♂ Moderate: 44% **Severe: 1%** |

**Note**:

* P stands for Percentile. Cells with highest and lowest values have been formatted in bold and italic, respectively.

For instance, while some studies report the mean of the items (scoring 1-7), others report the total score (scoring 20-140), and some others classify participants into different prevalence groups or severity levels. In this latter case, there is also a disparity between those studies that use numeric cut-off points to establish severity levels, and those that use clusters lacking a numeric cut-off point. This report-inconsistency may be due, among many other things, to the fact that the NMP-Q was not designed as a clinical questionnaire. Only the study of León-Mejía et al. [19] has explored clinical uses of this questionnaire, proposing specific cut-off points according to age and gender. Let us mention that this latter study was not included in this systematic review as it was published at the beginning of 2020, thus falling out of the time scope of this study.

When examining NMP classifications shown in Table 2, there is also a great disparity. However, most studies providing a scoring-system have differentiated between being "at-risk" of

developing nomophobia and "having" it. The problem arises when analyzing the latter since sometimes they were treated as a matter of "having vs. not having", and in some other cases it was just a question of having different levels of nomophobia (NMP, hereafter).

Out of eighty-two studies using the NMP-Q, sixty-five of them did not follow any classification system, whereas seventeen of them did it. Among the ones that provided a classification system, the 4-level proposed by Yildirim is the most common option.

Many times the presence of NMP is established when someone's score is above the average mean of items (and we see that most participants are in this situation) with a mean value that varies between 3.72 and 4.07. When looking at worrying levels of NMP in three or four-level-classifications, we find that between 25.7% and 67% have moderate problems, and between 1% and 87% have several problems of NMP.

When the studies report the mean of the total score, this goes from 45.5 to 76 (see Table 2 above in which cells with the highest and lowest values have been highlighted in red and blue, respectively).

Most studies follow a classification based on four levels of severity (absence, mild, moderate, and severe), but within this group there are also slight differences in the cut-off points used (see Table 2). Most of them put the value of 20 in the absence level, whereas others put it in the mild. Most studies place 100 as the cut-off point for severity, but for others it is 101. Moreover, most studies point to 140 as the maximum score for severity, whereas for others it is 120 or scoring above 100.

As for mild levels, most place it in the range 21-60, while for others it is between 16-20, 21-59, and 20-60. Also, most studies report the prevalence for the whole sample, but others provide it by gender and other grouping criteria. Some studies report the prevalence in all categories, whereas others only disclose moderate and severe cases. Finally, some studies provide the Mean and SD together with the severity information, while most do not.

As for those studies not using the NMP-Q (see Table 3), only eight out of twenty-six used a classification system, of which the most common is the 2-level classification ("at-risk" and "having NMP"). Here in this group, the range of "at-risk" goes from 27% to 81%, and "having NMP" goes between 18.5% and 73%.

**Table 3. Nomophobia classifications assessed with other tools.**

| LEVEL | STUDY | TOOL | CRITERIA | RESULTS |
|---|---|---|---|---|
| Having or not | Dongre et al. (2017) | ICD-10 | Meeting 3 or more criteria | Having NMP: 68.92% |
| 2-level CL | Dixit et al. (2010) | AHI | At risk: 0-24. Having NMP: 24 | *At risk: 81.5%* <br> **Having NMP: 18.5%** |
| 2-level CL | Bivin et al. (2013) | NSI-SR | At risk: 20-39. Having NMP: 40 > | At risk: 64% <br> Having NMP: 23% |
| 2-level CL | Pavithra et al. (2015) | AHI | At risk: 20-24. Having NMP: >24 | **At risk: 27%** <br> Having NMP: 39.5% |
| 2-level Cl | Prasad et al. (2017) | AHI | At risk: 34-39. Having NMP: ≤ 40 | At risk: 40.97% <br> Having NMP: 24.12% |
| 3-level CL | Sharma et al. (2015) | AHQS (5 questions) | Absence: 10-15. At risk: 16-28 <br> Having NMP: 29-40 | *Having NMP: 73%* |
| 4-level CL | Matoza & Carballo (2016) | RWT | Absence: 20. Mild: 20-60 <br> Moderate: 60-100 Severe: 100 | Mild: 43.6% <br> Moderated: 40.6% *Severe: 15.8%* |
| 4-level CL | Kar et al. (2017) | AHQS (9 questions) | Absence: 20. Mild: 16-20 <br> Moderate: 9-15; Severe: 9 | **Severe: 7.8%** |

**Note**:

AHI stands for Ad Hoc Instrument, and AHQs for Ad Hoc Questions

**Table 4. NMP as total score.**

| Prevalence as mean of the total score (oldest to newest study) | | |
|---|---|---|
| Chukwuemeka et al. (2017) | NMP-Q | 57.71 ± 16.69 |
| Chemara & Octaviani (2017) | NMP-Q | 53.7 ± 12.87 |
| Gezgin et al. (2017b) | NMP-Q | 79.71 ± 26,65 |
| Chandak et al. (2017)* | NMP-Q | 79.30 ± 13.82 |
| Yildiz-Durak (2018) | NMP-Q | 51.8 ± 1.29 |
| Aguilera-Manrique et al. (2018) | NMP-Q | 82.39 ± 18.63 |
| Al-Balhan et al. (2018)* | NMP-Q | 82.71 ± 22.68 |
| Yildiz (2019) | NMP-Q | **51.29 ± 26.26** |
| Blbüloğlu et al. (2019) | NMP-Q | 60.77 ± 15.09 |
| Lin et al. (2018) | NMP-Q | 74.65 ± 18.80 |
| Ahmed et al. (2019a) | NMP-Q | 81.45 ± 3.11 |
| Ahmed et al. (2019b) | NMP-Q | 77.6 ± 3.11 |
| Mean of total score split by groups | | |
| Jianling & Chang (2018)* | NMP-Q | ♀: 72.52 ± 24.22; ♂: 64.81 ± 23.57 |
| Gutiérrez-Puertas et al. (2019) | NMP-Q | In Spain: 78.84 ± 18.91; In Portugal: *93.82 ± 21.98* |

Moderate cases are around 40% and severe cases, between 7-15%. There is also one study that uses a reversed scoring, meaning the higher the score the lower the NMP. In this latter case, the four levels are as follows: Absence (above 20), Mild (16-20), Moderate (9-15), Severe (less than 9).

In Tables 2 and 3, we have reported the results of those studies that followed both a scoring and a classification system, but most studies just reported prevalence (their results are shown in Tables 4 and 5). In this group of studies, some of them provide the mean along with the level of severity, whereas others just report one piece of information.

In those studies using the arithmetic mean of items (Table 5) the score varies between 2.95 and 4.74. When NMP is reported as the mean of the total score (as Yildirim and his collaborators suggested) the value goes from 51.29 to 93.82 (Table 4).

**Table 5. NMP as mean of items.**

| Prevalence as mean of items (oldest to newest study) | | |
|---|---|---|
| Gezgin et al. (2017a) | NMP-Q | 3.96 |
| Gentina et al. (2018) | NMP-Q | 3.53 ±.92 |
| Gezgin et al. (2018a) | NMP-Q | 3.61 ±1.38 |
| Tams et al. 2018) | NMP-Q | **2.95 ±1.26** |
| Daei et al. (2019) | NMP-Q | 3.1 ±.72 |
| Kara et al. (2019) | NMP-Q | 3.10 ±.92 |
| Aktay & Hanife (2019) | NMP-Q | 4.19 ± 1.28 |
| Fitz et al. (2019) | NMP-Q | *4.74 ± 1.68* |
| Adnan & Gezgin (2016) | NMP-Q | 4.07 |
| Salwa (2017) | NMP-Q | 3.2 ±.97 |
| Gezgin et al. (2018c) | NMP-Q | 3.73 |

**Note**:

* They appear both in Tables 5 and 6 because they reported both the mean and severity levels.

** According to the authors, High and Low NMP Groups corresponds to the top 25 percent and bottom 25 percent, respectively.

**Table 6. Percentages and other ways of reporting prevalence.**

| Percentages and other ways of reporting NMP prevalence | | |
|---|---|---|
| Kaur et al.(2015) | Others | Absence: 15%; *At risk: 79%*; Having NMP: 6% |
| Yildirim et al. (2015) | NMP-Q | *Having NMP: 42.6%* |
| Farooqui & Pore (2016) | NMP-Q | Mild: 17.9%; Moderate: 60%; Severe: 22.1% |
| Tavolacci et al. (2015) | Others | Having NMP: 1 out of 3 |
| Han et al. (2017)l | NMP-Q | Low NMP group (N = 73); High NMP group (N = 74)** |
| Menezes & Pangam (2017) | Others | At risk: 64%; NMP: 26% |
| Kanmani et al. (2017) | NMP-Q | Absence: 1.2%; Mild: 41.6%; Moderate: 42%; Severe: 15.2% |
| Muralidhar et al. (2017) | NMP-Q | Absence: 3%; NMP: 97%; Mild: 33.3%; Moderate: 56.2%; Severe: 7.5% |
| Chandak et al. (2017)* | NMP-Q | NMP: 38% |
| Louragli et al. (2018) | NMP-Q | Moderate and Severe: 69.1% of girls and 63% of boys |
| Al-Balhan et al. (2018)* | NMP-Q | Mild: 18%; Moderate: 56.2%; Severe: 25.8% |
| Jianling & Chang (2018)* | NMP-Q | Absence: 17%; Low: 32.7%; Mild: 34%; Severe: 13.5%; Very severe: 2.5% |
| Sethia et al. (2018) | NMP-Q | Moderate: 61.5%; Severe: 6.1% |
| Aini et al. (2018) | NMP-Q | *Moderate: 73.3%*; High: 26,7% |
| Harish & Bharath (2018) | NMP-Q | NMP: 99%; Mild NMP: 36.1%; Moderate: 50.4%; Severe: 13.5% |
| Salinas et al. (2018) | Others | NMP: 37% |
| Bragazzi et al. (2019) | NMP-Q | Mild: 51.1%; Moderate: 41.4%; Severe: 7.4% |
| Jones et al. (2019) | NMP-Q | Triathlon: Mild; Polo athletes: Moderate |
| Veerapu et al. (2019) | NMP-Q | Mild: 17%; Moderate: 64.3%; Severe: 18.7% |
| Batool & Ayesha (2019) | NMP-Q | Mild: 5.3%; Moderate: 68%; *Severe: 26.7%* |
| Semerci (2019) | NMP-Q | Absence: 7.3%; Mild: 45.1%; Moderate: 39.5%; Severe: 8% |
| Jilisha et al. (2019) | NMP-Q | Absence: 9%; Mild: 20.8%; Moderate: 54.5%; Severe: 23.5% |
| Cain & Malcom (2019) | NMP-Q | Absence: 0.5%; Mild 24.5%; Moderate: 56.8%; Severe: 18.2% |
| Adawi et al. (2019) | NMP-Q | Mild: 51.1%; Moderate: 41.4%; Severe: 7.4% |
| Dasgupta et al. (2017) | NMP-Q | Engineering students: 44.6%; Medical students 42.6% |

**Note**:

* They appear both in Tables 5 and 6 because they reported both the mean and severity levels.

** According to the authors, High and Low NMP Groups corresponds to the top 25 percent and bottom 25 percent, respectively.

Some other studies report prevalence in terms of percentages of individuals who are nomophobic (Table 6).

As seen in Table 6, moderate cases go from 39.5% to 73.3%, and severe cases, from 6.1% to 26.7%. As for the percentages of "at-risk" subjects, these go from 64% to 79% whereas that subjects having NMP are between 6% to 42%. The mean percentage of nomophobic people in moderate cases is 54.7% whereas for severe cases is 16.1% (Fig 2).

## Gender and age differences

An important matter that has been discussed among the empirical studies on NMP is whether there are gender and age differences, and consequently, who are more affected by this phobia. A review of the current literature points to mixed results in both variables. We have screened all the reviewed studies and performed a descriptive analysis, counting and comparing the number of studies that found gender and age differences and those not finding differences at all or partial results.

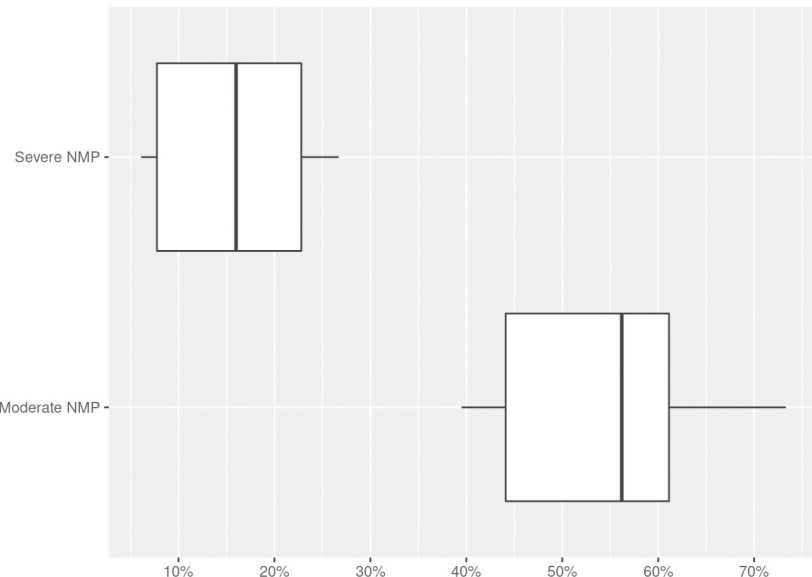

**Fig 2. Percentages of moderate and severe cases of NMP.**

## Gender differences

After searching for gender differences, we classified those studies reporting gender differences into two main groups: those finding females as more nomophobic and the opposite. Within these two groups, we also highlighted those with partial results.

The evidence tables show the results reported along with the country, instrument, and sample used by the study. We find that NMP is more prevalent among women since twenty-four studies found women to suffer more from NMP, while eight found the opposite (see Tables 7 and 8).

Notice that most of these studies finding women more nomophobic were conducted in Turkey and India, followed by other European, Asian and American countries. Therefore, all the places where these results were found are very culturally diverse suggesting that social factors may not explain this gender difference alone.

Also, there is a higher presence of undergraduate participants, and most participants are in their twenties.

Interestingly, the only study exploring genetic variables, with Turkish twins, also found females suffering more from NMP than males. As for studies finding more NMP in males, we have eight and three with partial results, and most of them were conducted in India, Turkey and Pakistan. Most of the participants are also students in their early twenties.

Our review shows that there is an evident report-inconsistency when dealing with gender differences, as some studies provide detailed data on which kind of differences were found, whereas others only report that gender differences were significant, sometimes not telling the direction of these differences, i.e., who are more affected, female o male participants.

Also, many studies do not address gender differences. When providing gender results, there are two different ways of showing this information: providing the percentages of females and males in the classification groups (at risk, nomophobic, mild or severe, etc.), and providing the nomophobia mean score of both females and males.

The fact that gender differences have been studied and reported with different criteria makes it difficult to reach sound conclusions. However, our analyses suggests that females

**Table 7. Studies pointing to females as more nomophobic (by year of publication, older to newest).**

| Study | Tool | Country | Sample | NMP more in females |
|---|---|---|---|---|
| Sharma et al. (2015) | Others | India | Undergraduates | ♀ |
| Tavolacci et al. (2015) | Others | France | Age: 20.0 ± 2.4<br>Undergraduates | ♀ |
| Yildirim et al. (2015) | NMP-Q | Turkey | Age: 20.02 ± 1.65<br>Undergraduates | ♀ |
| Uysal et al. (2016) | NMP-Q | Turkey | Students | ♀ |
| Gezgin & Çakır (2016) | NMP-Q | Turkey | High school students | ♀ |
| Gezgin et al. (2017a) | NMP-Q | Turkey | Pre-service teachers | ♀ |
| Salwa (2017) | NMP-Q | Saudi Arabia | Undergraduates | ♀ |
| Dasgupta et al. (2017) | NMP-Q | India | Age: 21.33 ± 2.36<br>Undergraduates | ♀ |
| Arpaci et al. (2017c) | NMP-Q | Turkey | Age: 21.94 ± 3.61<br>Undergraduates | ♀ |
| Arpaci et al. (2017a) | NMP-Q | Turkey | Age: 22.08 ± 3.73<br>Undergraduates | ♀ |
| Chandak et al. (2017) | NMP-Q | India | Residents, Teaching Hospital | ♀ |
| Kanmani et al. (2017) | NMP-Q | Turkey | Undergraduates workers | ♀* |
| González-Cabrera et al. (2017) | NMP-Q | Spain | Age: 15.41±1.22<br>Undergraduates | ♀ |
| Gezgin et al. (2018a) | NMP-Q | Turkey | High school students | ♀ |
| Peris-Hernï¿½ndez (2018) | ERA-RSI | Spain | Students | ♀ |
| Sethia et al. (2018) | NMP-Q | India | Students | ♀ |
| Jianling & Chang (2018) | NMP-Q | China | Age: 19.01 ± 1.23<br>Smartphone users | ♀ |
| Aguilera-Manrique et al. (2018) | NMP-Q | Spain | Age: 22.77 ± 3.65<br>Undergraduates | ♀ |
| Mallya et al. (2018) | NMP-Q | India | Undergraduates | ♀ |
| Yavuz et al. (2019) | NMP-Q | Turkey | High-school students | ♀ |
| Cain & Malcom (2019) | NMP-Q | USA | Undergraduates | ♀ |
| Aktay & Hanife (2019) | NMP-Q | Turkey | Undergraduates | ♀ |
| Deryakulu & Ursavaş (2019) | NMP-Q | Turkey | Age: 18.36 ± 6.71<br>Turkish twin-pairs | ♀ |
| Galhardo et al. (2020) | NMP-Q | Portugal | Age: 22.95 ± 5.36<br>Students | ♀ |
| **Partial results** | | | | |
| Prasad et al. (2017) | NMP-Q | India | Age: 21.99 ± 2.95<br>Undergraduates | ♀ More NMP<br>♂ More at risk |
| Yasan & Yildirim (2018) | NMP-Q | Turkey | Age = 22.45 ± 2.30<br>Undergraduates | ♀ NBATC and GUC ** |
| Gutiérrez-Puertas et al. (2019) | NMP-Q | Spain<br>Portugal | Age: 20.78 ± 3.16<br>Undergraduates | ♀*** |

**Note**:

* Females score more only in severe levels.

** Females score more in the dimensions of "Not Being able to Communicate" (NBATC) and "Giving Up Convenience" (GUC).

*** Gender differences are found in Portuguese but not in Spanish participants.

**Table 8. Studies pointing to males as more nomophobic (by year of publication, older to newest).**

| Study | Tool | Country | Sample | NMP Vulnerability |
|---|---|---|---|---|
| Pavithra et al. (2015) | Other | India | Age: 21.6 ± 3.1 Undergraduates and non-students | ♂ |
| Nagpal & Ramanpreet (2016) | NMP-Q | India | Undergraduates | ♂ |
| Matoza & Carballo (2016) | RWT | Paraguay | Age: 17-35 ± 21.9 Undergraduates | ♂ |
| Kar et al. (2017) | Other | India | Age: 21.08 Undergraduates | ♂ |
| Dongre et al. (2017) | ICD-10 | India | Age: 21.23 ± 9.44 Residents in an urban area | ♂ |
| Yildiz (2019) | NMP-Q | Turkey | High school students | ♂* |
| Jilisha et al. (2019) | NMP-Q | India | Undergraduates | ♂ |
| Daei et al. (2019) | NMP-Q | Iran | Undergraduates | ♂ |
| **Partial results** | | | | |
| Farooqui & Pore (2016) | NMP-Q | India | Undergraduates | ♂** |
| Nawaz et al. (2017) | NMP-Q | Pakistan | Smartphone users | ♂*** |
| Ozdemir et al. (2018) | NMP-Q | Pakistan and Turkey | Students | ♂**** |

**Note**:

* It is not statistically significant

** Females have more moderate levels but males have more severe levels

*** Females scored more in the dimension "Fear of Not Being Able to Access Information";

**** True for Turkey but not Pakistan.

seem to be more affected by NMP than males. Notice that we only included those studies that analyzed gender differences and also reported the direction of them.

## Age differences

When looking at the role of age, comparisons between studies are a bit more troublesome. Firstly, because the age groups were created with different age-points. Secondly, because age differences were studied regarding different parameters: having or not, suffering more or less, total scores, scores by NMP dimensions, among others. Thirdly, some studies analyzed the effect of age indirectly, for instance, differences in school grades between participants, being a freshman or residents vs. older undergraduates or graduates. This results in a myriad of age results difficult to read.

Despite this fact, we performed the same descriptive analyses as done with gender and presented this information in evidence Tables 9 and 10.

As shown, the number of studies finding younger participants to be more vulnerable to NMP is higher than those finding the opposite (9 vs 3, respectively). It is also higher than those pointing to mixed results or no age differences (9 vs 3 and 6, respectively).

Aside from that problematic factors that we explained above, some other issues claim for our attention if we are to better understand the role of age in the risk and prevalence of NMP. For instance, many studies have a narrow age range and this can affect the analyses. Moreover, NMP may be sensitive to gender, and having a very asymmetric gender distribution could also distort results. It is crucial to report the age range (youngest and oldest), which is not always done in studies that find significant age differences. Finally, some studies report the existence of significant age differences but fail to inform about the direction of these differences.

**Table 9. No age differences in NMP or mixed and partial results.**

| Study | Tool | Country | Sample | Results |
|---|---|---|---|---|
| **No age differences** | | | | |
| Yildirim et al. (2015) | NMP-Q | Turkey | Undergraduates. UGD: ♀ 74.6% Age: 17-34 (20.02 ± 1.65) | No statistically significant differences* |
| Gezgin & Çakır (2016) | NMP-Q | Turkey | Undergraduates. EGD Age: 20-24 | No significant differences in class level** |
| Gezgin et al. (2018a) | NMP-Q | Turkey | High school students. UGD: ♂ 60% | No differences in grades |
| Apak & Yaman (2019) | NMP-Q | Turkey | UGD: ♀ 60% | No differences |
| Gutiérrez-Puertas et al. (2019) | NMP-Q | Spain Portugal | Undergraduates (50% each country). UGD: ♀ 81% Age range: 17-39 (20.78 ± 3.16) | No statistically significant differences |
| Cain & Malcom (2019) | NMP-Q | U.S | Undergraduates. UGD: ♀ 64.6% Age range: under 26-66 | No differences |
| **Mixed results and NMP placed within a specific age range** | | | | |
| Chandak et al. (2017) | NMP-Q | India | Residents of teaching hospital. EGD Age range: 23-28 | More prevalence in ages 23-25 |
| Nawaz et al. 2017) | NMP-Q | Pakistan | Smartphone users. EGD Age range: 15-24 | FOGUC decreases with age***. FNBATAI increases with age**** |
| Sethia et al. (2018) | NMP-Q | India | Students. EGD Age range: 16-25 | Maximum scores in 20-22 years***** |

**Note**: UGD stands for Unequal Gender Distribution and EGD for Equal Gender Distribution

* The age comparison was in two groups: youngers (20 years or below) and elders (over 20 years).

** Differences examined in students of 9th to 12th grade.

***FOGUC stands for "Fear of Not Being Able to Access Information".

**** FNBATAI stands "Fear of Giving Up Convenience".

***** This maximum scores reflected moderate and severe levels.

When we look at age and gender differences (Fig 3), most of the studies do not examine these variables. When they do, these differences are not often interpretable (due to missing information) or statistically significant. We merged these two latter cases ("no information" and "not significant") in the category of "no results" and compare it with the categories "higher NMP" and "partial results" within four groups: females, males, younger, and older people. We see that "higher NMP" is more represented in females and young people, and therefore, they seem more vulnerable than males and older people. However, the fact that "no results" is the category most represented in the four groups and that partial results is similar distributed makes it difficult to establish any causality between age and gender, and the level of NMP (Fig 3).

## Common guidelines

One of the main conclusions of this systematic review is that heterogeneity in reporting data is exceptionally high. This fact, along with some inconsistencies highlighted above, makes it

**Table 10. Age differences.**

| Study | Tool | Country | Sample | Results |
|---|---|---|---|---|
| **NMP affecting more young people** | | | | |
| Kanmani et al. (2017) | NMP-Q | India | Undergraduates and working class. UGD: ♀ 60% Age range: 18-39 | NMP decreases with age |
| King et al. (2017) | IAT | Brasil | Clinical sample Age: 16-65 | More NMP in 18 to 29 |
| Gezgin et al. (2017a) | NMP-Q | Turkey | Pre-service teachers studying Education. UGD: ♀ 72% Age groups: <20, 20-22, 22>* | NMP decreases with age |
| González-Cabrera et al. (2017) | NMP-Q | Spain | Undergraduates. EGD Age: 13-19 (15.41±1.22). | NMP decreases with age |
| Peris et al. (2018) | ERA-RSI | Spain | Sample: Students. EGD Age range: 12-17 | NMP higher in ages 12-14 |
| Bernardini (2018) | Others | Italy | Young and early adults. EGD Age range: 18-36 | Mobile control increases with age |
| Blbüloğlu et al. (2019) | NMP-Q | Turkey | n = 360 Nurses. UGD ♀ 65.3% Age range: 18-47 | NMP decreased with age |
| Daei et al. (2019) | NMP-Q | Iran | n = 320 Age: 82.5% Students under 25. UGD: ♀ 59% | Age is significant ** |
| **NMP affecting more older people** | | | | |
| Matoza & Carballo (2016) | RWT | Paraguay | Undergraduates. UGD: 32.9% Age: 17-35 (M = 21.9) | Slight: 17-26 Moderate: 27-35 |
| Musa et al. (2017) | Others | Malaysia | Sample: Smartphone users. 20-30 (young) 30-40 (mature) | More NMP in the matured group |
| Jilisha et al. (2019) | NMP-Q | India | Undergraduates. UGD: ♀ 58.8% Age range: 22 | NMP higher in older participants |
| Yildiz (2019) | NMP-Q | Turkey | High school students Age: 12-18 | NMP increases with age. |

**Note**:

UGD stands for Unequal Gender Distribution and EGD for Equal Gender Distribution.

* Minimum and maximum age not specified.

** The authors only state that age has a significant relationship with NMP but not the direction.

extremely difficult to look at nomophobia prevalence, identify the magnitude of the problem, and shed light on gender and age differences. Therefore, we propose standard guidelines for future studies using the NMP-Q to move forward to the standardization of nomophobia reporting (see Fig 4).

Generating a common framework is fundamental for research purposes and produces high-quality data that can serve to inform and design therapeutic interventions. With this aim, and after reviewing carefully the studies published in ten years on NMP research, we propose

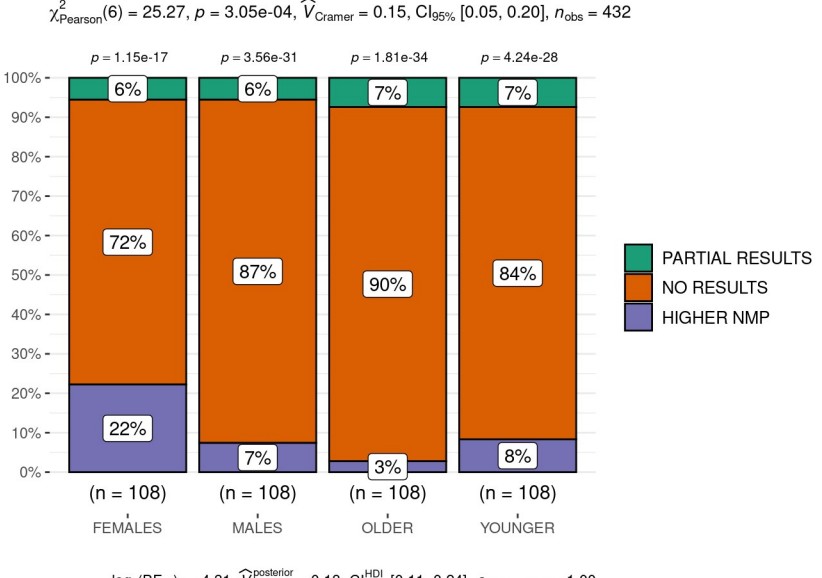

$\chi^2_{Pearson}(6) = 25.27$, $p = 3.05e\text{-}04$, $\widehat{V}_{Cramer} = 0.15$, $CI_{95\%}$ [0.05, 0.20], $n_{obs} = 432$

$\log_e(BF_{01}) = -4.31$, $\widehat{V}^{posterior}_{Cramer} = 0.18$, $CI^{HDI}_{95\%}$ [0.11, 0.24], $a_{Gunel\text{-}Dickey} = 1.00$

**Fig 3. Mosaic graph of gender and age differences.**

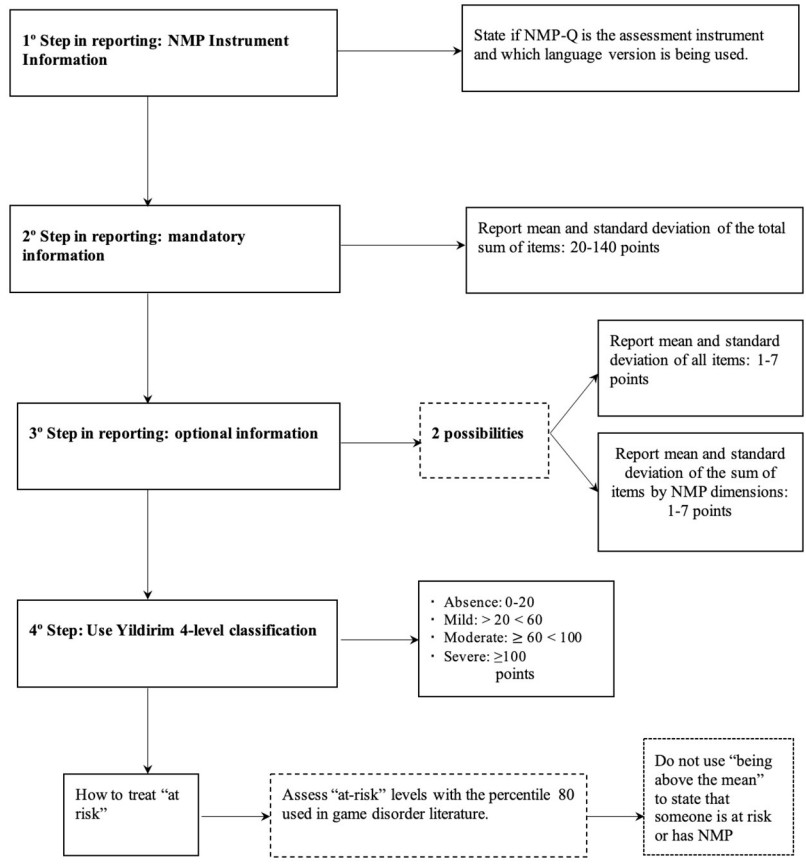

**Fig 4. Recommendations for reporting nomophobia prevalence.**

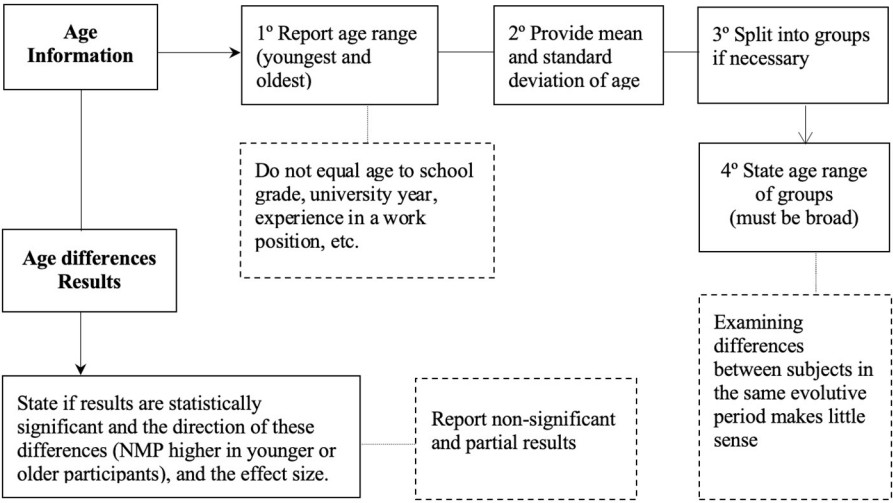

**Fig 5. Recommendations for reporting age information and results.**

the following steps and criteria. The most important information when reporting prevalence with the NMP-Q is the arithmetic mean and SD of the total sum of items: 20-140 points, since this is the way its author proposed. Reporting the same information by each dimension of the questionnaire or by item could be complementary to the total score, but it should never be used instead of it. The use of the levels of severity and cut-off points suggested by Yildirim [20] is also recommended, in addition to not treating this problem in a dichotomous way (non-suffering NMP vs Suffering NMP).

Providing the percentages of people in each level is welcome, and in case this is done, we suggest this information be reported in each of the four levels and not just in one or two. When trying to establish which people are "at-risk" of developing this problem, we do not recommend the criterion of being above the mean be followed. The reason is that it is neither theoretically nor methodologically supported. Instead, we can identify "at-risk" levels using the percentile 80 as done in the gaming disorder literature [21].

Studying the NMP prevalence by age is of great importance as younger participants seem to be at higher risk, but we need more systematic research to reinforce this conclusion (see Fig 5). The main problem is that age information is reported in very different ways, and good practices are not always followed.

We recommend providing the following basic information: age range of participants (total sample), age of the youngest and oldest, mean and SD. If participants are split into groups, it is also important to report this information along with the age range of the different groups. Bear in mind that if we want to analyze age and treat it as a relevant variable (which is highly recommended), the range must be sufficiently broad to distinguish wide evolutive periods or, at least, groups that differ in maturity or life events experienced. Also, we should not equal age to school grade, university year, or experience in a work position, which makes results difficult to apprehend and validate.

In case of reporting age differences, the significance should be explored as well as the direction of the differences found (NMP higher in younger or older participants). Also, reporting the effect size is recommended since is easy to calculate, and it gives us a more scientific approach by focusing on the size of the difference between two groups. Methods to analyse the effect size include Cohen's *d* (difference between two population means divided by their

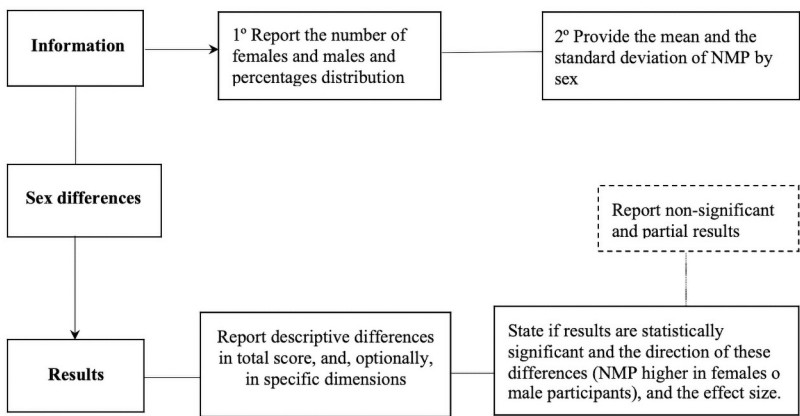

**Fig 6. Recommendations for reporting sex information and results.**

common standard deviation), Cohen's $f$ (the standard deviation of the population means divided by their common standard deviation) and for analyses of variance (ANOVAs) or covariance (ANCOVAs), and Eta Square ($\eta^2$). It is worth mentioning that Cohen's $d$ is the appropriate effect size measure if two groups have similar standard deviations and are of the same size. Glass's delta, which uses only the standard deviation of the control group, is an alternative measure if each group has a different standard deviation. And, finally, Hedge's g, which provides a measure of effect size weighted according to the relative size of each sample, is an alternative where there are different sample sizes. Non significant and partial results should be reported as this matters to the global comprehension of NMP.

Finally, for gender differences (see Fig 6) it is necessary to report the number of both females and males along with their percentage distribution, and the total mean and standard deviation of both genders, and additionally, differences or scores in specific items or dimensions of the NMP-Q. In case the significance of gender differences is examined, it is fundamental to inform of the direction of them (NMP higher in females or males), and to calculate the effect size (Cohen's $d$, Cohen's $f$ or $\eta^2$ as it improves substantially the quality of the analyses and makes it easier to conduct future meta-analyses on NMP. Non-significant and partial results should be reported as this matters to the global comprehension of this phobia.

Below we have proposed several recommendations for nonclinical research using the NMP-Q. For those interested in clinical implications, we recommend using the percentiles 15, 80, and 95 (a criterion widely used, for instance, in the literature of gaming disorders) that would refer to absence, at-risk, and NMP. There is just one work [19] that has identified cut off points for these percentiles, which are: 34 (P15), 72 (P80), and 94 (P95).

## Conclusions and limitations

The goal of this systematic review was to address the assessment and interpretation of nomophobia prevalence in scientific literature. First, we confirmed during the search, and final selection of papers, that there is some conceptual confusion in this field of research, as nomophobia is often equated with mobile addition despite these being two different (although related) constructs that emerged around the problematic use of smartphones. Second, there are many instruments to assess nomophobia, but the NMP-Q is clearly the most used worldwide, and its psychometric properties have been extensively studied. This, along with the fact that this questionnaire has been adapted into different languages, makes the NMP-Q the most

suitable instrument to conduct future studies and the only one allowing for transcultural research. In order to generalize results, it is crucial to calculate the total score of the NMP-Q and the percentages of individuals in the four-level classification, as suggested by Yildirim [20], and improving the size and representativeness of samples.

Since we identified many inconsistencies both when using this instrument and when reporting data, we have proposed some standard guidelines. This suggested protocol does not determine or limit the analyses to be done but rather, we conceive it as a helping instrument to build on a common and basic framework. We also recommended guidelines for gender and age analyses with the same supportive purpose that characterized the guidelines for reporting prevalence. In addition, we believe that calculating specific cut-off points by gender and age, and within different countries and cultural settings, will improve our understanding of the heterogeneity of findings coming from diverse countries. Our recommendation is to consider at least three percentiles (15, 80, and 95) when calculating those cut-off points.

Our review, with the evidence we have so far, confirms the existence of gender and age differences, pointing to females and young people as the most vulnerable groups, although this conclusion is based on a limited number of studies, a fact that weaken this statement. Therefore this initial result will need to be confirmed by future studies. Given this topic's clinical interest, we recommend analyzing age and gender differences as determining variables whenever it is possible and regardless of other primary goals. In doing so, we will contribute to identifying which groups need our attention and should be the target of interventions. In this sense, we make a plea to authors, reviewers, and editors to not consider partial or non-significant results as failure research or information with low value. Not publishing these results is neither positive for the clinical reasons explained above nor for guaranteeing that the published scholarly work is unbiased, ie., not rejected based on the direction and strength of findings [22]. This is not the only reason for this plea, since having a wider myriad of results will favor conducting meta-analyses on the subject.

One of our systematic review's main goals was to identify methodological inconsistencies when measuring and reporting nomophobia prevalence and generate practice, a field in which SR has proven to be useful. As some authors have highlighted, a SR should be seen as a means to an end, i.e., contributing to obtaining a robust and sensible answer to a research question, and not an end in itself [13, 23]. In this sense, we met our goals, but they are not exempt from limitations. First, SR has been widely used in the fields of experimental sciences for extracting information from control trials. Still, applying this research methodology to social science (an area to which belong most of the reviewed studies) is less common; it poses challenges when comparing studies that are very different in their designs, methodology, and results, and that sometimes describe poorly these sections. This precluded us from conducting any meta-analysis, and, in some cases, it was even difficult for us to compare descriptive results. For instance, the information on statistical significance and direction of significant differences was not always available. For this reason, and as said above, our finding pointing females and young people as more vulnerable to NMP should be taken with caution despite being based on the reviewed evidence. Our SR has not included studies from 2020, so other recent studies of potential relevance may not have been included in this review.

As for qualitative studies on NMP, our SR identified a small number of them that were excluded for not being related to our research and not meeting our inclusion criteria. It is true that whenever possible, the inclusion of qualitative studies helps to evaluate the effects of health interventions or to understand the experience of having a disease, to mention a few examples. Since our research question focused on prevalence and assessment data, our study could not benefit from the valuable perspective that qualitative designs can offer to any research. For all the above reasons, adapting the methodology of SR to nonmedical or nonexperimental fields,

it is only possible if one rejects a rigid approach and favor instead flexibility while sticking to the principles of rigor, transparency, and replicability [13] that we tried to follow despite the above limitations outlined.

It is clear that nomophobia has become a hot topic in the field of social sciences, particularly in cyberpsychology, and that we are concerned about the hours that high-school students and undergraduates spent with their mobiles and the sort of ties they establish with them. However, we noticed that there are substantially fewer studies done with children and preadolescents. Given that the age of first owning a mobile is decreasing, younger samples should be more targeted. For instance, research from other fields has found the association between digital technology use and well-being in adolescents to be negative but small [24]. Further studies on problematic use have reported that this problem affects one in every four children and young people, putting them in more danger of poorer mental health [13]. Whether called nomophobia or otherwise, the way we relate to our mobile phones has been linked to numerous psychological problems and different consequences for the individual's life [6, 25]. Therefore, Nomophobia research needs to contribute to this debate on the frequency, intensity, and harmful consequences of this phobia among the youngest.

To conclude, by improving our knowledge of nomophobia we will be guiding clinical and therapeutical studies, which were very salient when this topic emerged. Indeed, the field of intervention in NMP is underdeveloped and calls for collaborative work between scholars who have expert knowledge on this phobia and those who specialize in different groups of therapeutic frameworks within Behaviour Therapy. Many psychotherapists sometimes work outside the academy, not benefitting from all the existing research and theoretical knowledge. Therefore, we must end this gap and learn together how to help those already affected by this problem so they can enjoy a more balanced and healthier life. It is also crucial that nomophobia can be prevented through educational programs that instruct people in the appropriate use of technology. Currently, we are not aware of any programs that specifically address nomophobia with adolescents and other vulnerable groups. It would be very beneficial if educational interventions could be developed in the future to reduce nomophobic behaviour.

## Supporting information

**S1 Appendix.** A: Studies included in the qualitative synthesis. B: A selection of studies not included in the qualitative synthesis.
(PDF)

**S1 Checklist. PRISMA 2009 guidelines.**
(DOC)

## Author Contributions

**Conceptualization:** Ana C. León-Mejía, Mónica Gutiérrez-Ortega.

**Data curation:** Ana C. León-Mejía, Mónica Gutiérrez-Ortega.

**Formal analysis:** Ana C. León-Mejía, Mónica Gutiérrez-Ortega.

**Funding acquisition:** Joaquín González-Cabrera.

**Investigation:** Ana C. León-Mejía, Mónica Gutiérrez-Ortega.

**Methodology:** Ana C. León-Mejía, Mónica Gutiérrez-Ortega.

**Project administration:** Joaquín González-Cabrera.

**Supervision:** Ana C. León-Mejía.

**Validation:** Ana C. León-Mejía.

**Visualization:** Ana C. León-Mejía, Isabel Serrano-Pintado.

**Writing – original draft:** Ana C. León-Mejía.

**Writing – review & editing:** Ana C. León-Mejía, Mónica Gutiérrez-Ortega, Isabel Serrano-Pintado, Joaquín González-Cabrera.

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
