## [Decision Letter · Decision Letter 0]

8 Feb 2021

PONE-D-20-35929

A Systematic Review on Nomophobia Prevalence: Surfacing Results and Standard Guidelines for Future Research

PLOS ONE

Dear Dr. León Mejía,

Thank you for submitting your manuscript to PLOS ONE. After careful consideration, we feel that it has merit but does not fully meet PLOS ONE’s publication criteria as it currently stands. Therefore, we invite you to submit a revised version of the manuscript that addresses the points raised during the review process.

We look forward to receiving your revised manuscript.

Kind regards,

Giuseppe Carrà, PhD

Academic Editor

PLOS ONE

Journal Requirements:

2. “Please ensure that you include a title page within your main document. We do appreciate that you have a title page document uploaded as a separate file, however, as per our author guidelines (http://journals.plos.org/plosone/s/submission-guidelines#loc-title-page) we do require this to be part of the manuscript file itself and not uploaded separately.

3.We note that the grant information you provided in the ‘Funding Information’ and ‘Financial Disclosure’ sections do not match.

Reviewers' comments:

Reviewer's Responses to Questions

**Comments to the Author**

1. Is the manuscript technically sound, and do the data support the conclusions?

Reviewer #1: Yes

Reviewer #2: Partly

2. Has the statistical analysis been performed appropriately and rigorously? 

Reviewer #1: Yes

Reviewer #2: No

3. Have the authors made all data underlying the findings in their manuscript fully available?

Reviewer #1: Yes

Reviewer #2: Yes

4. Is the manuscript presented in an intelligible fashion and written in standard English?

Reviewer #1: Yes

Reviewer #2: Yes

5. Review Comments to the Author

Reviewer #1: In the current manuscript, Ana León Mejía and colleagues performed a systematic review on Nomophobia (No-Mobile-Phone-Phobia) prevalence, focusing their attention on the research instruments to assess this phenomenon, the prevalence criteria that researchers follow worldwide, and sex and age differences. Main findings suggest that females and young people seem to be more vulnerable to Nomophobia although methodological disparity makes it difficult to reach definitive conclusions.

The study is grounded on a solid scientific rationale and enquires a topical subject in psychiatry, as the concept of Nomophobia has been rising social concern since its introduction in 2008. Nevertheless, there is some conceptual confusion in this field of research and many inconsistencies both when using and reported data from the assessment tools. The paper stands out as an important contribution from both a research and a clinical perspective, also by recommending some common guidelines for future research.

The manuscript is consistent within itself and has a pleasant logic flow. The Introduction section is clear, providing the reader with exhaustive information needed to understand the article and contextualize the research question, which is clearly outlined and justified. The study design is suitable and described in a linear and comprehensive manner, and methods for data extraction are proper. The discussion has a pleasant logic flow and is adequately balanced between debating about the findings of the work and making more wide-ranging deductions and considerations. Implications for future research and clinical practice are well outlined. Tables display data in a precise and readable manner. References are appropriate.

I highly appreciated the Common guidelines section and the purpose to give the reader a global picture of the nomophobia research. Nonetheless, there are some minor issues that I wish the authors would fix to further improve the value of the article.

1. The protocol and the eligibility criteria, as well as the search strategy are well designed. However, finding out about unpublished studies and including their results in a systematic review when eligible and appropriate is important for minimizing bias. The standard advice is to search trial/study registries, as the WHO International Clinical Trials Registry Platform, ClinicalTrials.Gov, and the EU Clinical Trials Register, for information about unpublished or ongoing studies.

2. In the Methods section the authors should include a paragraph describing in a clearer and more accurate way the analyses conducted on the included studies (e.g., the results reported that females and young people seem to be more vulnerable to Nomophobia but in the Methods you did not explicit the analyses conducted to reach these findings). Moreover, I recommend declaring, if it is possible, how you assessed the risk of bias of the included studies (Critical Appraisal).

3. While Tables display data in a precise and readable manner, the contents of mosaic graph of age and gender differences, reported in Figure 2, could be better illustrated in the text above.

4. Strengths and future implications are well outlined. However, it could be useful to delineate the limitations in greater details in order to provide readers with additional suggestions for further research.

5. The Appendix could be enriched with an extensive list of each relevant study that was read in full-text form but excluded from the review.

Reviewer #2: I congratulate the authors on a timely and very important review. I have the following major comments

1) The research question needs to be defined so that there is clear link to harm. The outcome is Nomophobia in itself is not a problem if there is no clear link to harm. Smartphone addiction is indeed questionnable - see

Amy Orben & Andrew K. Przybylski

https://www.nature.com/articles/s41562-020-0840-y?proof=t

Therefore you need to include the literature that demonstrate where and harm may be caused by this. FPlease include this reference in the introduction to strengthen the argument

https://bmcpsychiatry.biomedcentral.com/articles/10.1186/s12888-019-2350-x

2) Methods: The authors exclude studies that do not include quantitative data. These should be included and the authors need to synthesis the evidence.

3) A key part of a SR is the Study Level Risk of bias. This needs to be included

4) Synthesis. The authors correctly argue that there is considerable heterogeneity across the included studies. They need to separate the studies out into subgroups that explain the heterogeneity. See The Cochrane handbook for systematic reviews, Section "Including Non randomised Studies" for guidance of how to carry out a systematic review of these types of studies. For example subgroup by study design, age, and sex

5) Consider a graphical representation of the prevalence data

6) Discussion. Revise after considering the possible public health interventions that may be possible for particular groups at highest risk of harm eg younger adults and females.

6. PLOS authors have the option to publish the peer review history of their article (what does this mean?). If published, this will include your full peer review and any attached files.

Reviewer #1: No

Reviewer #2: No

---

## [Author Response · Author response to Decision Letter 0]

7 Apr 2021

REVIEWER 1

First of all, we would like to thank you for your kind words to our study. As you may know, conducting a SR is resource-intensive and very time-consuming. Roughly, we spend more than two years doing this one. So, we appreciate your words and feel encouraged. Also, it is not that common that referees consider the positive aspects before highlighting the flaws. 

Regarding the minor issues that you would like us to address, we took care of them the best we could and tried to follow your advice as much as possible. For the sake of clarity, we will follow the same enumeration to tackle them:

1) Risk of bias

Your comment on publication bias and the opinion of reviewer 2 made us realize that we needed to explain this point better in a specific section. Indeed, it prompted a debate within our research team since bias in SR depends on many factors, e.g., choice of search criteria, literature sources, inclusion and exclusion criteria, etc. Dalton highlighted what he called "the hidden elephant of publication bias," defining it as the tendency to publish only statistically or clinically significant results, which is frequent in peer review journals. Publication bias also emerges when there is no discussion of inclusion criteria and PRISMA (not our case). In this sense, our study reviewed many studies that did not find significant results. Many of the journals come from India and Pakistan, which are not indexed in WoS or Scopus. Apart from WoS and Scopus, we searched Google Scholar, ProQuest, and Science Direct, looking for any study that met our inclusion criteria (prevalence/assessment data, peer review, focused on NMP, and written in English or Spanish). We only eliminated non-scientific publications such as newspaper articles and articles that were impossible to understand because they didn't have a minimum rigor). Also, notice that except for a couple of studies, the rest of the NMP studies reviewed did not follow an experimental methodology with trials, which is why we don't analyze effects and outcomes. Our samples are prevalence and correlational studies. Anyway, after reading your comments on trial registries, we checked them in case we missed something related to our research questions but didn't find it. This is the information regarding the risk of bias that we added to the article:

Risk of bias

No doubt, the existence of bias in SR is almost inevitable. Even though we tried to minimize it, there can always be some subjectivity in the screening process. In this SR, the two principal authors, who did the analyses, were in charge of this task and agreed 100% on the studies finally included, a process that is known to reduce the risk of bias considerably but not eliminate it (Mallet et al., 2012). Searching institutional websites is crucial to avoid publication bias, as research gathered in these repositories may contain relevant information (Dalton et al., 2017; Hedin et al., 2016). But it also has been said that these sources of information introduce other types of bias; for instance, differences in search functions across websites make it necessary to change or adapt the search strings (Mallet et al., 2012). Also, not all repositories are equally visible on the Internet or accessible to the researcher, and most of the debate is around unpublished trials not being represented in the SR. This latter fact does not affect this SR since the literature on NMP that we reviewed is not based on controlled trials but mostly on prevalence assessment and correlational designs. In any case, we tried to minimize bias by screening Google Scholar and other article repositories. Still, most of the publications on nomophobia that came out of the peer-reviewed channels did not meet all of our inclusion criteria (i.e., language, report prevalence/assessment data, and meeting a minimum of scientific standards of research). Those meeting our requirements were indeed included. Also, we analyzed a significant number of studies whose journals are not indexed in SCOPUS or WoS (mostly from India and Pakistan). Therefore, even though the possibility of missing pertinent studies is there, we followed all the steps to minimize it.

2) Clarify conclusions on gender and age differences

We are sorry if we could have been more explicative when concluding that females and young people are more nomophobic. Now we will try to explain this point better. Our analyses just organized the reported information of the studies reviewed into evidence tables, which are characteristics of systematic reviews. We counted descriptively how many studies found females more nomophobic (n=26) and the opposite (n=8), which is the information that tables 7 and 8 represents. We took into account total and partial differences. The fact that all countries were culturally and racially diverse and the fact that the only study exploring genetic variables with twins also found females more nomophobic reinforced the result of women seeming more vulnerable. As for age differences, we followed the same logic of counting and comparing studies reporting age differences. 

For the sake of clarity, and thanks to your observation, we added some pieces of text so our findings can be better understood by the reader (text in yellow):

Gender and age differences

An important matter that has been discussed among the empirical studies on NMP is whether there are gender and age differences, and consequently, who are more affected by this phobia. A review of the current literature points to mixed results in both variables. We have screened all the reviewed studies and performed a descriptive analysis, counting and comparing the number of studies that found gender and age differences and those not finding differences at all or partial results. 

Gender differences

After searching for gender differences, we classified those studies reporting gender differences into two main groups: those finding females as more nomophobic and the opposite. Within these two groups, we also highlighted those with partial results. The evidence tables show the results reported along with the country, instrument, and sample used by the study.

Age differences

When looking at the role of age, comparisons between studies are a bit more troublesome, firstly, because the age groups were created with different age-points. Secondly, because age differences were studied regarding different parameters: having or not, suffering more or less, total scores, scores by NMP dimensions, among others. Thirdly, some studies analyzed the effect of age indirectly, for instance, differences in school grades between participants, being a freshman or residents vs. older undergraduates or graduates. This results in a myriad of age results complicated to read. Despite this fact, we performed the same descriptive analyses as done with gender and presented this information in evidence tables 9 and 10. As shown, the number of studies finding younger participants to be more vulnerable to NMP is higher than those finding the opposite (9 vs 3, respectively). It is also higher than those pointing to mixed results or no age differences (9 vs 3 and 6, respectively).

3) To better illustrate the mosaic graph

Our intention behind including a mosaic graph was to help the reader see the big picture of gender and age differences. Perhaps we failed in explaining its value. We reformulated the text and introduced a new graph that we believe to be more illustrative and easier to read. 

When we look at age and gender differences (Fig. 3), most of the studies do not examine these variables. When they do, often these differences are not statistically significant or interpretable (due to missing information). We merged these two latter cases (“no information” and “not significant”) in the category of “no results” and compare it with the categories “higher NMP” and “partial results” within four groups: females, males, younger, and older people. We see that “higher NMP” is more represented in females and young people, and therefore, they seem more vulnerable than males and older people. However, the fact that “no results” is the category most represented in the four groups and that partial results is similar distributed makes it difficult to establish any causality between age and gender, and the level of NMP (Fig. 3).

4. Expand Limitations

You are right in that we did not address the existence of bias in our SR sufficiently and that we did not discuss the limitations of our study in much detail. We are very thankful for this observation. We added limitations and inserted them in the final remarks section (now renamed “Final remarks and limitations”), just between the penultimate and the last paragraph (see text in yellow).

Final remarks and limitations

(penultimate paragraph) We also recommended guidelines for sex and age analyses with the same supportive purpose that characterized the reporting prevalence guidelines. Also, we believe that calculating specific cut-off points by sex and age, and within different countries and cultural settings, will improve our understanding of the heterogeneity of findings from diverse countries. Our recommendation is to consider at least three percentiles (15, 80, and 95) when calculating those cut-off points.

→One of our systematic review's main goals was to identify methodological inconsistencies when measuring and reporting nomophobia prevalence and generate practice, a field in which SR has proven to be a useful tool. As some authors have highlighted, a SR should be seen as a means to an end, i.e., contributing to obtaining a robust and sensible answer to a narrowed research question, and not an end in itself (Lichtenstein et al.; 2008; Mallet et al., 2012). In this sense, we met our goals, but they are not exempt from limitations. First, SR has been widely used in the fields of experimental sciences for extracting information from control trials. Still, applying this research methodology to social science (an area to which belong most of the reviewed studies) is less common; it poses challenges when comparing studies that are very different in their designs, methodology, and results, and that sometimes describe poorly these sections. This precluded us from conducting any meta-analysis, and, in some cases, it was even difficult for us to compare descriptive results. For instance, the information on statistical significance and direction of significant differences was not always available. For this reason, and as said above, our finding pointing females and young people as more vulnerable to NMP should be taken with caution despite being based on the reviewed evidence. Our SR has not included studies from 2020, so other recent studies of potential relevance may not have been included in this review.

As for qualitative studies on NMP, we identified a small number of them that were excluded for not being related to our research and not meeting our inclusion criteria. It is true that whenever possible, the inclusion of qualitative studies helps to evaluate the effects of health interventions or to understand the experience of having a disease, to mention a few examples. Since our research question focused on prevalence and assessment data, our study could not benefit from the valuable perspective that qualitative designs can offer to any research.

→For all the above reasons, adapting the methodology of SR to nonmedical or nonexperimental fields, it is only possible if one rejects a rigid approach and favor instead flexibility while sticking to the principles of rigor, transparency, and replicability (Mallet et al., 2012) that we tried to follow despite the above limitations outlined.

To conclude, it is clear that nomophobia has become a hot topic in the field of social sciences, particularly in cyberpsychology….. (to the end of section)

5) Appendix

It is true that the Appendix could be enriched in many ways and that we could add those studies that were fully read but not included. The only problem is that the number of them is huge. A middle-ground solution would be including some of the more relevant ones that were not related to our research questions or did not meet the inclusion criteria. Also, this new appendix allows us to connect with some of the comments made by the reviewer 2.

REVIEWER 2

We genuinely thank the reviewer for her/his congratulations on the importance of the review and sincerely appreciate all the comments on our work that have improved the manuscript significantly. For the sake of clarity, we will follow the same enumeration to tackle them:

1) Redefine research question concerning harm and smart addiction

In the section “Protocol and eligibility criteria”, we stated 4 questions to conduct the SR. Consequently, this study was designed to address problematic issues related to the assessment and reporting of NMP results, which is a specific topic within the NMP literature. Since SR builds on previous questions, it is impossible to add, change, or eliminate none. Otherwise, it will require doing a completely different systematic search of articles and analysis. For this reason, we are very sorry for not being able to incorporate this comment.

Regarding the issue of "harm," we outlined the fundamental aspects of the complex construct of nomophobia in the introduction. Currently, there is consensus in defining nomophobia as related to the problematic use of the cell phone. Specifically, as an inability to control and regulate the use of the cell phone and suffer negative consequences in daily life. From many authors' point of view (including King, Yildirim and us), nomophobia is not understood as cell phone addiction but as a situational phobia. This view does not deny that NMP is related to other constructs or refuse to acknowledge the existence of negative consequences for people's lives, such as anxiety, panic disorder, depression, loneliness, etc. Indeed, all these problems are highlighted in the systematic review by Rodríguez-García et al. (2020) that we cited. What we mean is that a phobia is a clinical concept that already “entails harm”, in the form of major disruptions (that we explained in the introduction), and that is different (although related) from other nonclinical concepts as well as from the idea of mobile addiction. As for the two references, we sincerely thank the reviewer for this contribution that has enhanced our conclusions very much (see text in yellow):

To conclude, it is clear that nomophobia has become a hot topic in the field of social sciences, particularly in cyberpsychology, and that we are concerned about the hours that high-school students and undergraduates spent with their mobiles and the sort of ties they establish with them. However, we noticed that there are substantially fewer studies done with children and preadolescents. Given that the age of first owning a mobile is decreasing, younger samples should be more targeted. →For instance, research from other fields has found the association between digital technology use and well-being in adolescents to be negative but small (Orben & Przybylski, 2019). Further studies on problematic use have reported that this problem affects one in every four children and young people, putting them in more danger of poorer mental health (Sohn et al., 2019). Whether called nomophobia or otherwise, the way we relate to our mobile phones has been linked to numerous psychological problems and different consequences for the individual's life (Rodríguez-García et al., 2020; Sohn et al., 2019). Therefore, Nomophobia research needs to contribute to this debate on the frequency, intensity, and harmful consequences of this phobia among the youngest. 

2) Non-quantitative method 

We apologize for not being able to include your suggestions, although we share an interest in qualitative studies. Conducting a SR requires finding articles that serve to answer the research question. We were open to include any study analyzing prevalence and assessment of NMP, but we realized that only quantitative studies provided the information on assessment/prevalence that we were looking for. Indeed, during the searching period we identified very few qualitative studies on NMP that were excluded for this reason (not meeting inclusion criteria). In other words: they were excluded because they felt out of the scope: they did not focus on NMP prevalence, something mandatory due to our research questions. Therefore, and thanks to your suggestion, we added a note to the PRISMA figure: "No Quantitative" refers to studies not providing any prevalence/assessment data and not to the research methodology

Also, and due to a comment of reviewer 1, we added another appendix with studies fully read but not included in the review, among which several of them are qualitative research. 

3) Risk of bias

You are very right. It is true that we discussed bias very superficially and did not have a specific section for that. Many thanks for your observation that we fully incorporated. Following your comment and the words of reviewer 1, we have added a bias section and expanded the limitations too, which now can be found in the renamed section “final remarks and limitations."

Risk of bias

No doubt, the existence of bias in SR is almost inevitable. Even though we tried to minimize it, there can always be some subjectivity in the screening process. In this SR, the two principal authors, who did the analyses, were in charge of this task and agreed 100% on the studies finally included, a process that is known to reduce the risk of bias considerably but not eliminate it completely (Mallet et al., 2012). Searching institutional websites is crucial to avoid publication bias, as research gathered in these repositories may contain relevant information (Dalton, et al., 2017; Hedin, et al., 2016). But it also has been said that these sources of information introduce other types of bias; for instance, differences in search functions across websites make it necessary to change or adapt the search strings (Mallet et al., 2012). Also, not all repositories are equally visible on the Internet or accessible to the researcher, and most of the debate is around unpublished trials not being represented in the SR. This latter fact does not affect this SR since the literature on NMP that we reviewed is not based on controlled trials but mostly on prevalence assessment and correlational designs. In any case, we tried to minimize bias by screening Google Scholar and other article repositories. Still, most of the publications on nomophobia that came out of the peer-reviewed channels did not meet all of our inclusion criteria (i.e., report prevalence/assessment data and meeting a minimum of scientific standards of research). Those meeting our requirements were indeed included. Also, we analyzed a significant number of studies whose journals are not indexed in SCOPUS or WoS (mostly from India and Pakistan). Therefore, even though the possibility of missing pertinent studies is there, we followed all the steps to minimize it.

5. Final remarks and limitations

→One of our systematic review's main goals was to identify methodological inconsistencies when measuring and reporting nomophobia prevalence and generate practice, a field in which SR has proven to be an effective tool. As some authors have highlighted, a SR should be seen as a means to an end, i.e., contributing to obtaining a robust and sensible answer to a narrowed research question, and not an end in itself (Lichtenstein et al., 2008; Mallet et al., 2012). In this sense, we met our goals, but they are not exempt from limitations. First, SR has been widely used in experimental sciences to extract information from control trials. Still, applying this research methodology to social science (an area to which belong most of the reviewed studies) is less common; it poses challenges when comparing studies that are very different in their designs, methodology, and results, and that sometimes describe poorly these sections. This precluded us from conducting any meta-analysis, and, in some cases, it was even difficult for us to compare descriptive results. For instance, the information on statistical significance and direction of significant differences was not always available. For this reason, and as said above, our finding pointing females and young people as more vulnerable to NMP should be taken with caution despite being based on the reviewed evidence. Our SR has not included studies from 2020, so other recent studies of potential relevance may not have been included in this review.

As for qualitative studies on NMP, we identified a small number of them that were excluded for not being related to our research and not meeting our inclusion criteria. It is true that whenever possible, the inclusion of qualitative studies helps to evaluate the effects of health interventions or to understand the experience of having a disease, to mention a few examples. Since our research question focused on prevalence and assessment data, our study could not benefit from the valuable perspective that qualitative designs can offer in research.

→For all the above reasons, adapting the methodology of SR to nonmedical or nonexperimental fields, it is only possible if one rejects a rigid approach and favor instead flexibility while sticking to the principles of rigor, transparency, and replicability (Mallet et al., 2012) that we tried to follow despite the above limitations outlined.

4) Synthesis and heterogeneity

The studies reviewed in this SR are not based on controlled trials. Therefore, the Cochrane guidelines on how to include non-randomized studies do not apply to our research. The heterogeneity mentioned in the article is related to the various instruments used to assess NMP, the different classification criteria followed, and mixed ways of reporting prevalence results. For this reason, and when examining NMP prevalence, we presented different evidence tables that classify the studies according to similarity in these 3 main variables (instrument, criteria and reporting). 

5) Graphical representation of the prevalence

We thank you for this great observation that led to a new graph representing the percentages of moderate and severe cases found by the reviewed studies (See Figure 2. Percentages of moderate and severe cases of NMP). As shown throughout the article, the variety of forms reporting NMP prevalence is enormous. For this reason, it would be impossible to include all of them in just one graph (that is why we showed this information in different evidence tables). But continuing with the harmful component of nomophobia that you flagged up, we decided to display the moderate and severe cases in a boxplot. 

6) Adding interventions

We find the field of clinical interventions full of possibilities and fascinating. For this reason, we would love to contribute to this in the future. However, this SR focused on prevalence and assessment problems. And from our results on these specific issues, we proposed practical guidelines for future research. Adding intervention guidelines too would considerably expand the scope and length of this article. We genuinely believe that this requires another piece of work entirely devoted to this task, as doing it superficially may not add much value. However, in the final section, we mentioned the need for enhancing our knowledge of NMP to develop clinical and therapeutical lines of action. After reading your comment, we decided to reinforce this message and adding the following bit:

By improving our knowledge of nomophobia, we will be guiding clinical and therapeutical studies, which were very salient when this topic emerged. → Indeed, the field of intervention in NMP is underdeveloped and calls for collaborative work between scholars who have expert knowledge on this phobia and those who specialize in different groups of therapeutic frameworks within Behaviour Therapy. Many psychotherapists sometimes work outside the academy, not benefitting from all the existing research and theoretical knowledge. Therefore, we must end this gap and learn together how to help those already affected by the dark consequences of mobile phone ties to enjoy a more balanced and healthier life. It is also crucial that nomophobia can be prevented through educational programs that instruct people in the appropriate use of technology. Currently, we are not aware of any programs that specifically address nomophobia with adolescents and other vulnerable groups. It would be very beneficial if educational interventions could be developed in the future to reduce nomophobic behaviour.

---

## [Editor Report · Decision Letter 1]

8 Apr 2021

A systematic review on nomophobia prevalence: Surfacing results and standard guidelines for future research

PONE-D-20-35929R1

Dear Dr. León Mejía,

We’re pleased to inform you that your manuscript has been judged scientifically suitable for publication and will be formally accepted for publication once it meets all outstanding technical requirements.

Kind regards,

Giuseppe Carrà, PhD

Academic Editor

PLOS ONE

---

## [Editor Report · Acceptance letter]

22 Apr 2021

PONE-D-20-35929R1 

A systematic review on nomophobia prevalence: Surfacing results and standard guidelines for future research 

Dear Dr. León Mejía:

I'm pleased to inform you that your manuscript has been deemed suitable for publication in PLOS ONE. Congratulations! Your manuscript is now with our production department. 

Kind regards, 

on behalf of

Dr. Giuseppe Carrà 

Academic Editor

PLOS ONE